# Advancing Theorem Proving in LLMs through Large-Scale Synthetic Data

**Huajian Xin**[3*]  **Daya Guo**[1]  **Zhihong Shao**[1]  **Z.Z. Ren**[1]  **Bo Liu**[1]
**Chong Ruan**[1]  **Wenda Li**[3]  **Xiaodan Liang**[2,4†]
[1]DeepSeek  [2]Sun Yat-sen University  [3]University of Edinburgh  [4]MBZUAI
H.Xin-3@sms.ed.ac.uk, {guoday, zhihongshao, rzz, zhuqh, chong.ruan}@deepseek.com,
benjaminliu.eecs@gmail.com, wli8@ed.ac.uk, xdliang328@gmail.com

## Abstract

Proof assistants like Lean have revolutionized mathematical proof verification by providing high levels of accuracy and reliability. Although large language models (LLMs) have demonstrated potential in mathematical reasoning, their advancement in formal theorem proving is hindered by the scarcity of large, high-quality training datasets. To address this challenge, we present a novel approach to generate extensive Lean 4 proof data from natural language mathematical problems at the high school and undergraduate levels. Specifically, we synthesize 8 million formal statements with corresponding proofs, leveraging this dataset to fine-tune the DeepSeekMath 7B model. The resulting model, DeepSeek-Prover, achieves a pass rate of 50% on the Lean 4 miniF2F benchmark, surpassing the previous state-of-the-art result of 41.0%. These findings underscore the potential of large-scale synthetic data in significantly enhancing the theorem-proving capabilities of LLMs.

## 1  Introduction

Formal mathematical languages, such as Lean [Moura and Ullrich, 2021], Isabelle [Paulson, 1994], and Coq [The Coq Development Team], have enabled the development of computer-verifiable proofs [Avigad, 2023]. However, constructing formal proofs remains a labor-intensive process that requires both substantial effort and specialized expertise—often challenging even for experienced mathematicians. To ease the burden of writing formal proofs, recent approaches [Polu and Sutskever, 2020, Jiang et al., 2021, Han et al., 2021, Polu et al., 2022, Lample et al., 2022, Jiang et al., 2022, Yang et al., 2023] have explored the use of language models to automatically generate proofs for given formal statements. Despite this progress, the performance of these methods has been constrained by the limited availability of high-quality formal proof data.

To address this limitation, we propose an iterative methodology for generating large-scale Lean 4 proof data from natural language mathematical problems. Our approach utilizes large language models to translate competition-level problems from high school and undergraduate mathematics into formal statements, followed by the generation of verifiable proofs using the Lean 4 prover. The model is fine-tuned with this synthetic data, and the data generation process is repeated to further refine the model. This iterative pipeline, executed with the continuously enhanced DeepSeekMath 7B model [Shao et al., 2024], continues until no further improvement in model performance is observed. Ultimately, our final dataset comprises 8 million formal statements paired with corresponding proofs.

---

*Work done during the internship at DeepSeek.
†Corresponding author.

38th Conference on Neural Information Processing Systems (NeurIPS 2024).

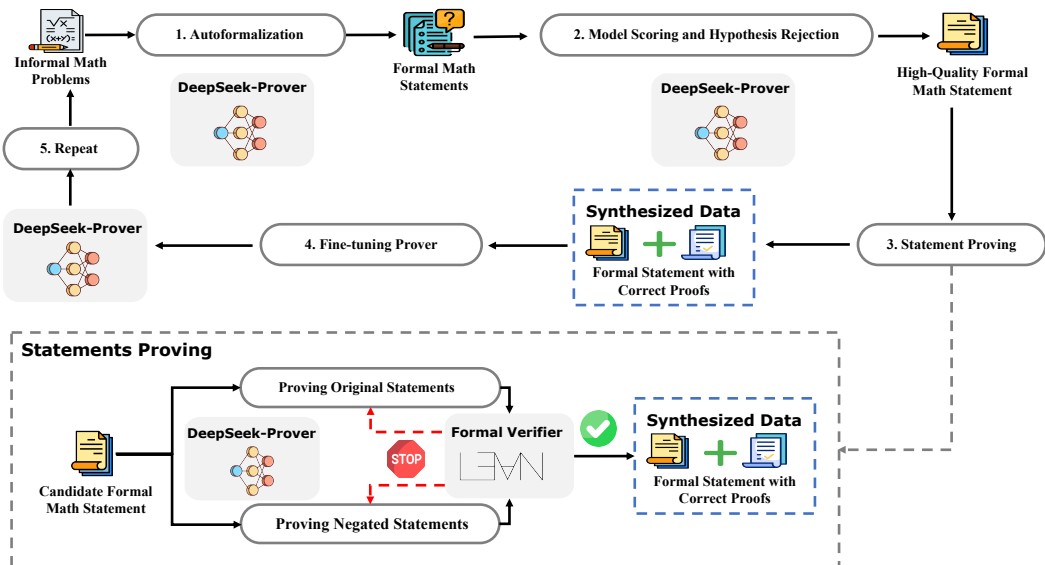

Figure 1: **Overview of our approach.** Our pipeline starts by autoformalizing a broad range of mathematical problems into formal statements. Inconsistent or overly simplistic statements are discarded. The remaining high-quality statements are then iteratively processed, attempting either to prove or refute them (i.e., prove their negation), until one is verified by the Lean 4 prover. The verified pairs of natural language problems and their formalization, formal statements (or their negations) and their corresponding proofs are used to fine-tune the model, enhancing its theorem-proving ability. This process repeats until no further performance gains are observed.

We evaluate the theorem proving performance in Lean 4 of the resulting model, DeepSeek-Prover, using the miniF2F benchmark [Zheng et al., 2022]. DeepSeek-Prover achieves a pass rate of 50% on the test set of the miniF2F, surpassing the previous state-of-the-art result of 41.0%. Ablation studies reveal that our iterative training process progressively enhances the model's problem-solving ability with each iteration, further demonstrating the effectiveness of our approach.

## 2 Approach

Our approach consists of an iterative cycle of dataset synthesis and model training, as shown in Figure 1. Each phase is described in detail below.

**Informal Data Curation.** We collect mathematical problems in natural language by scraping online resources containing high school and undergraduate exercises, exams, and competitions. After cleaning the data, we curated a dataset of 869,659 high-quality math problems, focusing on algebra and number theory.

**Model Initialization.** The model, initialized from DeepSeekMath-Base 7B [Shao et al., 2024], is fine-tuned on MMA dataset [Jiang et al., 2023], which includes formal statements from Mathlib, the standard mathematical library for Lean 4. This enhances the model's basic autoformalization capabilities. Additionally, we include theorem-proving data from LeanDojo [Yang et al., 2023], adapted from a next-tactic prediction task to full-proof generation. We refer to the resulting model and its further improved versions as DeepSeek-Prover.

**Model Scoring and Hypothesis Rejection.** Initially, many autoformalized statements were low quality. To improve this, we introduced a scoring mechanism that prompts DeepSeek-Prover to assess each statement using a chain-of-thought approach. Statements are categorized as "excellent," "good," "above average," "fair," or "poor." Statements rated "fair" or "poor" are discarded. Additionally, some provable statements contained inconsistent hypotheses leading to vacuous conclusions. To filter these, DeepSeek-Prover attempts to prove the statement with `False` as the conclusion. Successful proofs of

these transformed statements reveal inconsistent hypotheses, and these statements are discarded. This process refines the dataset, leaving 712,073 high-quality formal statements for proof generation.

**Statement Proving.**   With a large corpus of high-quality formal statements, DeepSeek-Prover attempts to generate proofs. Since some synthesized statements may be semantically incorrect and therefore unprovable, brute-force search would be inefficient. To optimize this, we exploit the logical symmetry between a statement and its negation. Dual concurrent proof searches are initiated for each statement—one for $\Gamma \vdash P$ and another for $\Gamma \vdash \neg P$. DeepSeek-Prover samples up to $k$ times for each statement until a valid proof for either is found. All validated proofs, whether for the statement or its negation, are used to further train DeepSeek-Prover. This method enriches the dataset with both propositions and their negations, even if the original propositions were incorrectly formalized.

**Iterative Enhancement.**   Since the pipeline relies on DeepSeek-Prover, improving its performance after each iteration is crucial. Verified pairs of formal statements (or their negations) and their corresponding proofs are used to enhance the model's theorem-proving capabilities. Pairs of natural language problems and their formalized counterparts, when correctly proved, are also collected to improve the model's autoformalization abilities. With each cycle of refinement, DeepSeek-Prover's performance in both autoformalization and theorem proving incrementally improves, contributing to better dataset synthesis. This iterative process continues until no further performance gains are observed.

# 3   Experiments

## 3.1   Main Results

**Benchmark and metric.**   We assess the theorem-proving capabilities of DeepSeek-Prover using the miniF2F benchmark [Zheng et al., 2022], which comprises 244 problems ranging from elementary arithmetic to competition-level challenges, including those from the American Invitational Mathematics Examination (AIME), the American Mathematics Competitions (AMC), and the International Mathematical Olympiad (IMO). Our experiments are based on the Lean 4 version of miniF2F, as provided by the LeanDojo project [Yang et al., 2023]. The primary evaluation metric is pass@$K$, which indicates the model's ability to generate a correct proof within $K$ attempts.

**Evaluation results.**   Table 1 presents a comparison of various theorem-proving methods on the miniF2F-test dataset. DeepSeek-Prover achieves a pass rate of 50.0% with a $64 \times 1024$ sampling budget, significantly surpassing the previous state-of-the-art pass rate of 41.0% achieved by Hypertree Proof Search [Lample et al., 2022] with $64 \times 5000$ tree search steps. Notably, DeepSeek-Prover achieves a considerable pass rate of 46.3% with only 128 sampling steps, surpassing most previous methods with few computational resources allocated. These results demonstrate DeepSeek-Prover's robustness and its ability to tackle complex proofs even under varying resource constraints.

| Method | Sampling Budget | miniF2F-test |
|---|---|---|
| COPRA (Code Llama) [13] | 500 | 5.7% |
| COPRA (GPT-3.5) [13] | 60 | 9.0% |
| COPRA (GPT-4) [13] | 60 | 26.6% |
| Llemma-7B [2] | 3200 | 26.2% |
| Llemma-34B [2] | 3200 | 25.8% |
| ReProver [16] | - | 26.5% |
| LLMStep [15] | 3200 | 27.9% |
| GPT-f [11] | $64 \times 4096$ | 36.6% |
| Hypertree Proof Search [7] | $64 \times 5000$ | 41.0% |
| DeepSeekMath-Base [12] | 128 | 19.67% |
| | 128 | 46.3% |
| DeepSeek-Prover (Ours) | $64 \times 128$ | 48.8% |
| | $64 \times 1024$ | **50.0%** |

Table 1: Comparison of state-of-the-art methods on the miniF2F-test dataset.

## 3.2 Ablation Studies

We performed ablation studies to evaluate the contributions of different components of DeepSeek-Prover, using pass@128 as the evaluation metric on the miniF2F-test dataset. The results are summarized in Table 2.

| Model | #Tokens | miniF2F-test |
|---|---|---|
| - | - | 27.5% |
| Mathlib | 0.2B | 31.2% |
| Synthetic Data | 3.1B | 42.6% |

(a)

| Scored Class | miniF2F-test |
|---|---|
| "fair" and "poor" | 38.1% |
| "excellent", "good" and "above average" | 42.6% |

(b)

| Iteration | miniF2F-test |
|---|---|
| 0 | 34.0% |
| 1 | 39.3% |
| 2 | 41.4% |
| 3 | 45.1% |
| 4 | 46.3% |

(c)

| Dataset Size | miniF2F-test |
|---|---|
| 1,000 | 24.18% |
| 10,000 | 31.97% |
| 100,000 | 37.7% |
| 1,000,000 | 38.11% |
| 8,066,621 | 40.16% |

(d)

Table 2: Ablation studies of data synthesis and model training components.

**Effectiveness of Large-Scale Autoformalization.** We conducted a comparative analysis between our autoformalized synthetic dataset and the human-written Mathlib dataset (the standard mathematical library of Lean 4), as shown in Table 2a. The models trained on our synthetic data substantially outperformed those trained solely on Mathlib data. This process employed an expert iteration strategy [Polu and Sutskever, 2020], where formal proofs were iteratively generated and used to fine-tune the model until performance improvements plateaued.

**Effectiveness of Formal Statement Scoring.** We evaluated the impact of formal statement quality on model performance by training on both high- and low-scoring proofs. As shown in Table 2b, models trained on high-score proofs outperformed those trained on low-score proofs by 4.5%. This result highlights the importance of accurate statement scoring for filtering lower-quality statements and improving overall performance.

**Effectiveness of Iterative Enhancement.** The results in Table 2c show a clear correlation between the number of iterations in data synthesis and improved theorem-proving performance. Each iteration refines the model's ability to handle increasingly complex proofs, resulting in significant performance gains. This iterative enhancement approach contributes to the generation of higher-quality synthetic data and bolsters the model's theorem-proving capabilities.

**Effectiveness of Scaling Synthetic Theorem-Proving Data.** As illustrated in Table 2d, there is a clear relationship between the size of the synthetic dataset and the model's performance on the miniF2F benchmark. Performance increases consistently with the size of the dataset, highlighting the critical role of large-scale data in advancing automated theorem proving. This underscores the necessity of systematic, large-scale data generation for further progress in the field.

## 4 Conclusion

In this paper, we introduced a method for generating extensive synthetic proof data from high-school and undergraduate-level mathematical competition problems. This approach significantly improved the performance of the DeepSeekMath 7B model in automated theorem proving (ATP) when trained on this synthetic dataset. Our model surpasses all previous state-of-the-art methods on the miniF2F-test benchmark for theorem proving in Lean 4. While our current work primarily focuses on algebra and number theory problems at the middle school and undergraduate levels, future work will aim to

broaden the scope of mathematical domains, enhancing the general applicability of our approach to a wider range of theorem proving tasks.

## Broader Impact

The research presented in this paper has the potential to significantly advance automated theorem proving through the use of large-scale synthetic proof data generated from informal mathematical problems. This progress can facilitate the development of tools for formalizing mathematical reasoning, supporting the broader mathematical and educational communities.

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
