# OpenReview forum: "Advancing Theorem Proving in LLMs through Large-Scale Synthetic Data"
_NeurIPS.cc/2024/Workshop/MATH-AI — MATH-AI 24_

### Official Review · Reviewer_pMFU · 2024-09-27
**A Good Iterative Pipeline for Autoformalization and Formal Theorem Proving**

**Rating:** 7
**Confidence:** 5

**Review:**

This paper introduces DS-prover, an iterative pipeline that employs LLMs to autoformalize natural language problems, filters high-quality autoformalizations, and fine-tunes LLMs for formal theorem proving in Lean. The proved statements generated by the LLMs are then added to the dataset to train the LLMs for autoformalization further. Experiments on miniF2F demonstrate that large-scale formal data derived from natural problems can significantly enhance the theorem-proving capabilities of LLMs.

The paper is well-written and easy to follow. Leveraging large-scale informal data to improve formal theorem proving is both intuitive and practical. While the high-level idea of an iterative pipeline is not novel and has been adopted in various existing works, this paper contributes by providing extensive experiments that convincingly demonstrate the effectiveness of their method.

---

### Official Review · Reviewer_pqsA · 2024-10-02
**Use data synthesis and model training cycles to improve theorem proving**

**Rating:** 7
**Confidence:** 5

**Review:**

This paper is technically solid and very complete for a workshop submission. The general structure of writing makes sense (There are still some typos in this paper though. I would suggest the authors to polish their writing a bit. E.g. line 37, "cycle" v.s. "circle", "model" v.s. "mode".). The method is detailed clearly. Using a cycle of data synthesis plus model training sounds promising, and is along the same line as most state-of-the-art works in the field such as AlphaProof. Sources of model initialization and data curation sound reasonable too. Overall, the method design and evaluation are in good quality. I would recommend acceptance of this work, and would be curious to see it scaled up.

---

### Official Review · Reviewer_T5tT · 2024-10-03
**Great work on the creation and usage of synthetic data for formal theorem proving**

**Rating:** 9
**Confidence:** 4

**Review:**

### Summary:

This work shows a method for effectively utilizing synthetic data for formal theorem proving by providing a pipeline to autoformalize informal proofs, maintain a high quality of synthetic data, and iteratively fine-tune models to achieve state-of-the-art results.

### Strengths:

This work achieves state-of-the-art results on miniF2F by a significant margin.

The ablation studies are comprehensive and show the effectiveness of each part of the methodology.

Data scarcity is a significant problem for formal theorem proving, and this work shows an effective method for using synthetic data in this domain.

### Limitations:

The resources (compute/time) needed to replicate the work are not shown.

---

### Decision · Program_Chairs · 2024-10-07

Accept